# CC Chemokines in Idiopathic Pulmonary Fibrosis: Pathogenic Role and Therapeutic Potential

**DOI:** 10.3390/biom13020333

**Published:** 2023-02-09

**Authors:** Shanshan Liu, Chang Liu, Qianrong Wang, Suosi Liu, Jiali Min

**Affiliations:** 1National Clinical Research Center for Metabolic Diseases, Key Laboratory of Diabetes Immunology, Ministry of Education, Department of Metabolism and Endocrinology, The Second Xiangya Hospital of Central South University, Changsha 410011, China; 2Drug Clinical Trial Institution, Children’s Hospital, Capital Institute of Pediatrics, Beĳing 100020, China

**Keywords:** idiopathic pulmonary fibrosis, CC chemokines, fibroblasts, targeted therapy

## Abstract

Idiopathic pulmonary fibrosis (IPF), characterized by progressive worsening of dyspnea and irreversible decline in lung function, is a chronic and progressive respiratory disease with a poor prognosis. Chronic or repeated lung injury results in inflammation and an excessive injury-repairing response that drives the development of IPF. A number of studies have shown that the development and progression of IPF are associated with dysregulated expression of several chemokines and chemokine receptors, several of which have been used as predictors of IPF outcome. Chemokines of the CC family play significant roles in exacerbating IPF progression by immune cell attraction or fibroblast activation. Modulating levels of detrimental CC chemokines and interrupting the corresponding transduction axis by neutralizing antibodies or antagonists are potential treatment options for IPF. Here, we review the roles of different CC chemokines in the pathogenesis of IPF, and their potential use as biomarkers or therapeutic targets.

## 1. Introduction

Idiopathic pulmonary fibrosis (IPF) is a chronic, progressive respiratory disease characterized by excessive extracellular matrix (ECM) deposition, interstitial fibrotic lesions, and architectural distortion. Patients with IPF experience pulmonary function decline and progressive worsening of dyspnea; prognosis is poor, with death typically occurring within 3 to 5 years [1]. Usual interstitial pneumonia (UIP) is regarded as the histopathological pattern of this devastating disease [2]. However, treatments aimed at minimizing inflammation have failed or have even had deleterious effects on IPF, suggesting that IPF may be more a fibrotic than inflammatory disease. Currently, there is no available curative treatment for IPF. Two therapies (nintedanib and pirfenidone), which have been approved for treatment of IPF, are only effective at slowing disease progression. According to existing investigations, cigarette smoking [3], viral infection [4], gene mutation [5,6] and other factors are common risk factors associated with the incidence of IPF, but the precise etiology of this fibrotic disease remains elusive.

When the pulmonary epithelium or endothelium is impaired by various insults, the coagulation pathway is triggered to prevent blood loss, followed by acute inflammation and activation of the innate immune system [7]. Innate immune cells in the alveolar microenvironment, including resident macrophages, neutrophils and dendritic cells, cooperate with cytokines to activate the adaptive immune response [7]. The chemoattractants released induce migration of immune cells to the site of injury, where they attempt to eliminate the initial insults to restore homeostasis. Abnormal tissue repair following epithelium impairment and failed elimination of associated insults may ultimately lead to fibrotic foci and irreversible lung respiratory function decline. As the final pathological outcome of many chronic inflammatory diseases, fibrosis is closely related to dysregulated inflammation and excessive deposition of collagen. Chemokines involved in the above pathological processes play crucial roles in IPF pathogenesis by acting as mediators of cellular communication.

Chemokines (chemotactic cytokines, 8–10 kDa) comprise a family of small soluble proteins originally discovered as mediators of directional migration of immune cells to sites of inflammation and injury [8,9]. These secreted proteins are subclassified into four main subfamilies based on the configuration of the two cysteine residues closest to the N-terminus: CC chemokines (two adjacent cysteines near the N-terminus), CXC chemokines (two cysteines separated by one other amino acid), CX3C chemokines (two cysteines separated by three amino acids) and XC chemokines (containing a single N-terminal cysteine) [8]. Chemokines exert their biological effects by binding to cell surface receptors that are selectively expressed on the surfaces of target cells. Classical chemokine receptors are proteins with a seven transmembrane structure and couple with G protein receptors, including CC, CXC, CX3C and XC chemokine receptors (CCRs, CXCRs, CX3CRs and XCRs, respectively), for signal transduction within cells [8,10].

Studies of CX3C chemokines and XC chemokines in IPF have rarely been reported. However, accumulating evidence has shown that the development and progression of IPF are associated with the dysregulated expression of CC chemokines and CXC chemokines in lung tissues, suggesting that these chemokines may be involved in its pathogenesis [11]. Some chemokines mainly function as chemoattractants to recruit leukocytes to sites of lung injury; some chemokines can also activate cells to initiate an immune response or promote wound healing. In this review, we analyze the role and diverse mechanisms of the most well-known chemokine ligands (CCLs), related CCRs and the role of the CCL-CCR transduction axis in IPF pathogenesis, highlighting potential CC chemokine-dependent treatment options for IPF. The cellular sources, target cells, and biological functions of reported CCLs in IPF are described in the following sections and summarized in Figure 1.

## 2. CC Chemokines and Receptors in IPF

### 2.1. CCL1

CCL1, also known as I-309 in humans and TCA-3 in mice, is a small glycoprotein secreted by activated T lymphocytes, endothelial cells and activated monocytes. By interacting with the cell surface chemokine receptor CCR8, CCL1 has a chemotactic effect on Th2 cells, monocytes, natural killer (NK) cells, immature B cells and dendritic cells [12,13,14,15]. CCL1 is involved in the pathogenesis of many chronic pulmonary diseases. CCL1 is a highly specific biomarker with great accuracy for lung adenocarcinoma diagnosis using bronchoalveolar lavage fluid (BALF) specimens [16]. In chronic obstructive pulmonary disease (COPD) patients, CCL1 gene variants are associated with acute exacerbations (AEs), which are a major cause of morbidity and mortality in this chronic disease [17]. In addition, the level of CCL1 is elevated in asthmatic airways, and neutralization of CCL1 or deficiency of its receptor CCR8 results in decreased mucosal lung inflammation and airway hyperresponsiveness, revealing the crucial role of CCL1 in bronchial asthma and other disorders characterized by inappropriate mast cell activation [18]. CCL1 is upregulated in human mesenchymal stem cell (hMSC)-induced PF in the murine sclerodermatous graft-versus-host disease (Scl-GVHD) model. Administration of a CCL1-blocking antibody alleviates the severity of this kind of lung fibrosis by suppressing infiltration of inflammatory cells into the lungs [19].

Our recent work showed that expression of CCL1 is enhanced in lung tissues of PF patients and mice. CCL1 exerts its fibrogenic activity via interaction with autocrine motility factor receptor (AMFR) but not its classical receptor CCR8. This CCL1-AMFR interaction on the membrane of fibroblasts causes PKCα-mediated AMFR phosphorylation, through which AMFR acquires E3 ligase activity, followed by ubiquitination and translocation of the endogenous ERK inhibitor Spry1 to the plasma membrane. At the plasma membrane, Spry1 binds to RasGAP, relieving the inhibitory effect of Spry1 on Ras-ERK-p70S6K signaling activity and enhancing profibrotic protein synthesis in fibroblasts. Genetic and pharmacological inhibition of the CCL1 signaling pathway has potent therapeutic efficacy against PF. Our study revealed a mechanism through which CCL1 influences PF development, suggesting this chemokine as a potential target for treating fibroproliferative lung diseases [20].

### 2.2. CCL2

CCL2, also named monocyte chemoattractant protein-1 (MCP-1), exhibits chemotactic activity for monocytes, microglia, T cells, NK cells and fibrocytes by binding to the receptor CCR2 [21,22,23]. A large number of studies have confirmed the role of CCL2 in IPF. Augmented CCL2 expression is found in IPF patients [24,25] and correlates negatively with the carbon monoxide diffusing capacity of the lung (DLco) and arterial oxygen tension [26]. Serum levels of CCL2 are associated with macrophage activation, and its upregulation increases the mortality risk of IPF patients [27]. In addition, CCL2 levels in BALF in IPF are significantly higher than those in other types of interstitial lung disease (ILD), including interstitial pneumonia collagen vascular disease (IP-CVD) and chronic interstitial pneumonia (CIP). Measurement of CCL2 levels in both BALF and serum may be helpful for discriminating IPF from other types of ILD. Although CCL2 levels in both BALF and serum are elevated in patients with different types of ILD compared with healthy volunteers, only patients with IPF exhibit significantly higher BALF CCL2 levels than serum CCL2 levels [24], which enabled us to distinguish IPF from other types of ILD [24,27]. Due to the correlation between serum CCL2 level and IPF progression, CCL2 can be used as a serum marker to predict the clinical course of IPF [24]. Supporting the critical role of CCL2-CCR2 signaling in IPF, gene knockout of CCL2 or its receptor CCR2 protects mice from bleomycin-induced PF [28,29,30].

Numerous studies have shown that CCL2 contributes to IPF through a variety of mechanisms. Apart from its major role as a chemoattractant of monocytes for immunological surveillance, new findings suggest its functional role in upregulation of ECM proteins. Indeed, CCL2 promotes upregulation of endogenous transforming growth factor beta 1 (TGF-β1) in pulmonary fibroblasts and stimulates collagen synthesis indirectly via autocrine and/or juxtacrine loops [31]. Stimulation of IPF patient-derived fibroblasts with CCL2 directly increases levels of α-smooth muscle actin (α-SMA) and procollagen [32], suggesting that fibroblasts, which act as the effector cells in IPF, are another target cell affected by CCL2 (Figure 2). Furthermore, fibrotic fibroblasts isolated from IPF patients are hyperresponsive to CCL2 compared with those from healthy individuals. CCR2, a receptor of several ligands (including CCL2), has been reported to be involved in IPF by acting on immune cells. Inflammatory monocytes and interstitial macrophages expressing CCR2 are active in patients with IPF [33]. CCR2 deficiency improves IPF outcome by attenuating macrophage infiltration and production of macrophage-derived matrix metalloproteinases [34]. Although CCR2 signaling has been shown to exert a profibrotic role [29], pulmonary CCR2+ CD4+ T cells are reported to attenuate PF development [35], revealing that CCR2 expression may have different effects on distinct immune cell subsets in the process of fibrosis.

CCL2 is expressed by various cell types. Strong expression of CCL2 has been detected on pulmonary alveolar epithelial cells (AECs) from IPF patients but not non-IPF subjects [36]. In addition to epithelial cells, macrophages, endothelial cells, and smooth muscle cells in IPF lung specimens express CCL2. However, these cells in non-IPF lung specimens also express CCL2 [37]. Hence, the enhanced CCL2 expression in IPF patients may derive from AECs [36]. Similar observations were found in BLM-induced fibrotic mice, in which CCL2 was upregulated in the activated epithelium in fibrotic areas [37], further supporting the notion that AECs are prominently affected by CCL2 during PF. A recent study further identified the novel mechanism by which diverse injuries promote CCL2/12 expression by AECs [38], showing that activation of the mTOR signaling pathway leads to AEC production of CCL2/12 and suggesting that targeting this pathway to reduce CCL2 production in AECs may be a viable antifibrotic strategy.

In addition to AECs, pulmonary fibroblasts are another primary source of elevated CCL2 in IPF [31,38]. Pulmonary fibroblasts isolated from patients with IPF produce significantly higher levels of CCL2 than nonfibrotic lung fibroblasts [39]. In fibroblasts, nuclear factor-κB (NF-κB) and activator protein-1 (AP-1) are responsible for transcriptional expression of CCL2 in IPF [39]. The high levels of thymic stromal lymphopoietin (TSLP) and TSLP receptor (TSLPR) in IPF have been shown to induce release of CCL2 in fibroblasts, which further contributes to PF by inducing subsequent monocyte chemotaxis [40]. STAT3 is confirmed to be required for TSLP-induced CCL2 expression. Moreover, stimulation of macrophages with macrophage colony-stimulating factor (M-CSF) [30], interleukin-13 (IL-13) [32] or collagen type I [41] also causes enhancement of CCL2 expression and perpetuates fibrotic responses in IPF. Collectively, the elevated CCL2 levels occurring in IPF derive from different cell types in response to various factors.

Bleomycin (BLM) injury increases CCL2 production in the lung by up to twofold, but it is reduced to basal levels after regulatory T (Treg) cell adoptive transfer therapy [42]. The inhibitory effect of amniotic fluid stem cells (AFSCs) on progression of BLM-induced PF is attributed to a reduction in CCL2 in BALF [43]. The tyrosine kinase inhibitor nintedanib (NTD), a drug approved for IPF treatment, can also reduce CCL2 production [44]. Carlumab, a human antibody neutralizing CCL2 [45], failed to produce a benefit in patients with IPF in a phase 2 trial. The most likely reason for its failure is that compensatory mechanisms become overstimulated in response to CCL2 inhibition [45]. Due to the significant enhancement of CCL2 in the serum and BALF of IPF patients and its critical role in IPF development, further studies are needed to identify new agents that can effectively suppress CCL2, which may provide significant benefits to patients with progressive IPF.

### 2.3. CCL3 and CCL4

CCL3 (macrophage inflammatory protein (MIP)-1α) and CCL4 (MIP-1β), proinflammatory chemokines secreted by activated T cells, B cells, macrophages, monocytes, NK cells, neutrophils, epithelial cells, fibroblasts, and osteoblasts, induce recruitment of dendritic cells, neutrophils, monocytes, macrophages, NK cells and T cells to inflammatory sites. CCL3 is a ligand for the receptors CCR1 and CCR5; CCL4 is a ligand for CCR5 but has low affinity for CCR1. CCR5 is recognized as a shared cellular receptor in activated T lymphocytes and alveolar macrophages (AMs) [46]. CCL3 preferentially mediates chemotaxis of CD8+ T cells, whereas CCL4 predominantly induces migration of activated CD4+ T cells [47].

Several groups have shown involvement of CCL3 and CCL4 in the pathogenesis of IPF. Compared with that in the lungs of healthy controls, CCL3 and CCL4 expression is increased in the lungs of IPF patients and different animal models, including the BLM model, silica model and mustard gas model. CCL3 and CCL4 have been shown to be involved in the inflammatory process of IPF. CCL3 levels in IPF patients correlate significantly with both the percentage and total number of neutrophils and eosinophils. Although Bless et al. [48] observed that CCL4 contributes to recruitment of neutrophils in rats, no correlation was found between CCL4 levels and the number or percentage of neutrophils in IPF patients. Moreover, no significant correlations were found between concentrations of CCL3 and CCL4 in BALF and lung function in patients with IPF [20]. In addition, CCL3 levels in the BALF of surviving and nonsurviving IPF patients do not differ significantly [49]. Although these mediators have been shown to contribute to the process of PF, their expression levels are not associated with the outcome of IPF patients.

In a murine model of BLM-induced lung fibrosis, the time-dependent increase in CCL3 after a BLM challenge was associated with an increased number of leukocytes and elevated profibrotic factors [50,51]. Targeting CCL3 by treatment with anti-CCL3 antibodies is beneficial for relieving BLM-induced lung injury and subsequent PF [50,51]. The accumulation of collagen and increased numbers of intrapulmonary macrophages and fibroblasts in BLM-induced fibrosis are attenuated in CCL3-/- and CCR5-/- mice but not in CCR1-/- mice, demonstrating that locally produced CCL3 participates in BLM-induced immune cell recruitment and subsequent development of lung fibrosis via interactions with CCR5 [52]. Compared with the number of studies on CCL3, relevant in vivo studies of CCL4 in lung fibrosis are quite limited. Nonetheless, existing studies confirm the potential profibrotic roles of CCL3 in the inflammatory mechanism of IPF, as well as promising therapeutic options for this devastating disease. Determining the intracellular mechanism downstream of the CCL3-CCR5 axis in target cells may provide a better understanding of IPF and possible approaches for its treatment.

### 2.4. CCL5

CCL5, also known as RANTES, is widely expressed in a variety of immune and nonimmune cells. The most studied receptor of CCL5 is CCR5, but it also binds to CCR1, CCR3, CCR4, CD44 and GPR75. Interaction between CCL5 and its receptor CCR5 induces recruitment of different leukocyte types, including T cells, monocytes/macrophages, dendritic cells, eosinophils and basophils, to sites of injury.

It has been reported that expression of CCL5 mRNA in BALF cells and levels of CCL5 protein in BALF are significantly increased in patients with distinct ILD compared with healthy volunteers [53]. Elevated CCL5 in IPF patients appears to be involved in accumulation of inflammatory cells in the lung. CCL5 released by lung epithelial cells induces recruitment of eosinophils into the lung, a characteristic feature of pulmonary disorders such as IPF [54]. Upregulated CCL5 in the lungs of patients with IPF is accompanied by increased recovery of lymphocytes or eosinophils in BALF [53]. All of these results suggest that CCL5 contributes to inflammatory cell accumulation in the lung during PF.

CCR5, the most studied interacting partner of CCL5, is expressed on various cell types, including T cells, macrophages, dendritic cells, eosinophils, and microglia. It is also the major coreceptor for human immunodeficiency virus (HIV) cell entry [46]. Fibroblasts from the lobes of patients with UIP (also referred to as IPF if the disease is idiopathic) exhibit strong expression of CCR5 [55]. Apart from fibroblasts derived from lobes, AMs and lymphocytes in BALF also express CCR5. In contrast to the similar expression level of CCR5 in AMs between IPF patients and controls, CCR5 expression is significantly reduced in lymphocytes from patients with IPF compared with those from controls [21]. Such a difference in expression of CCR5 in fibrotic lung disease emphasizes the potential importance of CCL5 in IPF pathogenesis. In addition to CCL5, other high-affinity ligands that bind to CCR5 are CCL3 and CCL4. As mentioned above, CCL3 and CCL4 are involved in IPF progression. Therefore, targeting CCR5, which results in inhibition of the profibrotic roles of these three chemokines, may be an effective strategy for IPF.

Although CCL5 expression is upregulated in patients with IPF, this characteristic is not unique to IPF and occurs in other types of ILD, including sarcoidosis and IP-CVD. Although measurement of CCL5 expression in BALF cells is inadequate for distinguishing IPF from other ILDs, such data provide new evidence that CCL5 is involved in pulmonary fibrotic disease and that modulation of CCL5 is a potential therapeutic strategy for IPF. Although several studies have indicated that CCL5 participates in the inflammatory process of IPF [56], the exact role of CCL5 in the process of fibrosis remains obscure, and further studies are required to illustrate the potential mechanism of the CCL5 axis in IPF.

### 2.5. CCL7

CCL7 (monocyte chemoattractant protein-3, MCP-3), first characterized from osteosarcoma supernatant, is a member of the MCP subfamily and also includes CCL2 and CCL8. CCL7 exhibits the broadest range in that it activates lymphocytes, monocytes, dendritic cells, NK cells, eosinophils, basophils and neutrophils [57]. CCR1, CCR2 and CCR3 are widely acknowledged as the main functional receptors for CCL7 [58], and several studies have reported that CCL7 can bind to CCR5 and CCR10 [59,60]. CCL7 is expressed at significantly higher levels in UIP lung biopsies than in biopsies from patients with other ILDs, and is localized to interstitial areas of the lung [50], suggesting that CCL7 may exert effects on inflammation-associated remodeling events in fibrotic disease. Higher levels of CCL7 are found in cultures of idiopathic pulmonary pneumonia (IIP) fibroblasts than in those of non-IIP fibroblasts, and CCL5, a CCR5 agonist, significantly increases synthesis of CCL7 in UIP fibroblasts [55]. Although CCL7 has been demonstrated to bind CCR5 with high affinity, it cannot elicit a functional response [61]. Hence, high levels of CCL7 may act as a natural antagonist of CCR5 by affecting its binding with other ligands. Similar to CCL21-induced CCL5 enhancement [62], the concomitant inflammatory chemokine and chemokine receptor profiles of CCL7 indicate mutual impacts caused by diverse chemokines in the context of IPF. The potential role of CCL7 in progression of IPF remains unclear, and further exploration of the mechanism of action of CCL7 and its main source may enable a broader understanding of this inflammation-associated disease.

### 2.6. CCL8

As a member of the MCP subfamily, CCL8 (monocyte chemotactic protein, MCP-2) plays key roles in allergic and inflammatory responses by attracting diverse immune cells, including eosinophils, basophils, mast cells, monocytes, T cells and NK cells. It is produced by many cell types and signals through interactions with CCR1, CCR2, CCR3, CCR5 and CCR8 [57,63].

A recent study reported CCL8 involvement in IPF development. A global transcriptome analysis revealed that CCL8 expression is higher in fibroblasts from IPF patients than in fibroblasts from controls [64]. Subsequent ontology and pathway analyses revealed that CCL8 is involved in essential pathways associated with IPF progression, including pathways related to the extracellular region, receptor binding, and chemokine activity [64]. In line with these results, researchers have confirmed that CCL8 protein levels are much higher in the BALF of patients with IPF than in that of healthy cohorts [64]. The increase in CCL8 in the BALF of IPF patients suggests its clinical relevance as a candidate marker for diagnosis and prognosis of IPF [64]. A cutoff value of 2.29 pg/mL has been demonstrated to show a high degree of accuracy for diagnosis, and IPF subjects with CCL8 levels > 28.61 pg/mL exhibit a decreased survival rate [64]. Although the precise influence and relevant mechanism of CCL8 in IPF pathogenesis remain elusive, the utility of this chemokine as a candidate molecule for differential diagnosis and prediction of survival in the IPF context provides insights into the immunoregulatory features of this irreversible disease.

### 2.7. CCL17 and CCL22

CCL17 (thymus and activation-regulated chemokine, TARC) and CCL22 (macrophage-derived chemokine, MDC) are T helper type 2 (Th2)-type chemokines that share 37% identity at the amino acid level. CCR4, their shared receptor, is highly expressed in macrophages, dendritic cells, NK cells, monocytes, platelets, neurons, microglia and functionally distinct T-cell subsets, including activated T cells, Th2 cells and Treg cells. As development of IPF is a Th2-mediated process, there is growing evidence that CCL17, CCL22 and CCR4 are involved in its pathogenesis [65,66,67,68]. CCL17 and CCL22 expression, along with that of their shared receptor CCR4, is significantly elevated in a rat model of radiation pneumonitis/PF and a BLM-induced PF model [69,70]. Moreover, both chemokines are upregulated in the BALF and serum of IPF patients [69,71,72]. Based on peripheral blood proteomic profiling data of IPF biomarkers in the multicenter IPF-PRO Registry, circulating levels of CCL17 and CCL22 are higher in the IPF patients than in controls [71]. Overall, elevated levels of CCL17 and CCL22 in BALF are possible prognostic markers of poor outcome in patients with IPF [49], supporting the putative significance of these two chemokines in IPF.

Several studies have revealed the underlying mechanisms of CCL17, CCL22 and CCR4 in IPF development. Yurika Yogo et al. reported localization of CCL22 and CCR4 at CD68+ AMs, and an inverse correlation was observed between CCL22 levels in BALF and DLco/alveolar ventilation per minute (VA) values in IPF patients [72], suggesting that CCL22 might induce lung dysfunction in IPF patients by recruiting and activating CCR4+ AMs [72]. In a BLM-induced mouse PF model, lethal inflammatory and fibrotic responses were provoked in wild-type mice but not seen in CCR4-knockout (CCR4-/-) mice [73]. The AMs and bone-marrow-derived macrophages isolated from CCR4-/- mice did not exhibit CCL17-dependent M1 activation but showed an M2 phenotype with higher expression of the nonsignaling CC chemokine scavenging receptor D6 in cell membranes [73], suggesting that scavenging receptor D6 acts as a novel component in regulating CCL17-mediated macrophage function and that CCL17-dependent activation of CCR4 in macrophages plays critical roles in the development of BLM-induced PF [73]. CCR4-deficient macrophages transiently revert to an M2 phenotype, providing a protective effect during the inflammatory response and suggesting that targeting CCR4 may be an effective approach to provide protection against IPF [73].

Apart from macrophages, BAL CD4 T cells show higher expression levels of CCR4 in IPF patients than in controls [74,75]. Despite such enhanced expression of CCR4 and elevated proportion of lung CD4+ T cells in IPF patients, the increase in the ratio of CCR4+ CD4 T cells to CCR6+ CD4 T cells in IPF correlates significantly with better lung function [76]. This finding suggests that specific subsets of cells may be protective in IPF and that augmentation of chemokines to recruit protective T-cell subsets may be a novel approach for IPF therapy [76]. Inconsistent with the conventional detrimental roles of CCR4, CCL17 and CCL22 in IPF, the study indicated a positive correlation between CCR4 expression and preservation of lung function in IPF patients. Due to the complex inflammatory mechanisms and etiology of IPF, the specific role of the CCL17/CCL22-CCR4 axis in the disease requires further exploration.

### 2.8. CCL18

CCL18 was originally discovered in the lung, and CCL18 expression was found to be induced in alternatively activated macrophages (AAMs); as such, CCL18 was initially named pulmonary and activation-regulated chemokine (PARC), alternative macrophage activation-associated CC chemokine-1 (AMAC-1), dendritic cell chemokine 1 (DC-CK1), and macrophage inflammatory protein-4 (MIP-4) [77,78]. CCL18 is known to bind to two receptors, CCR8 and PITPNM3, to exert effects on target cells [79,80].

Markedly enhanced CCL18 expression in the serum and BALF of IPF patients has been reported [81,82]. Prasse, A. et al. identified CCL18 in serum as the first biomarker to explicitly predict mortality in IPF [83]. In post hoc assessment of that trial, CCL18 in blood, and not other baseline biomarkers such as CCL13 and CXCL13, was the most consistent predictor of disease progression across IPF cohorts, suggesting its potential to inform new target discovery and clinical trial design [84]. A close negative correlation between serum CCL18 concentration and pulmonary functions was observed in patients with fibrotic lung diseases [85]. In addition, single nucleotide polymorphisms (SNPs) in the CCL18 gene, which influence CCL18 mRNA and protein expression, may predispose patients with IPF towards an unfavorable prognosis [86].

However, according to a case–controlled study by Ganesh Raghu et al., there is no significant association between CCL18 concentration at baseline and the rate of disease progression in IPF patents, indicating that the baseline value of CCL18 cannot predict disease progression [87]. In contrast to the observations of Prasse et al. [88], CCL18 was not found to have value in predicting IPF progression in another study, and no significant difference in baseline concentration of CCL18 between progressors and nonprogressors was found [87]. The inconsistency of these reports may be attributed to the study design because Prasse et al. characterized a monocentric cohort, whereas the other research was multicentric and multinational, with geographic factors contributing most to the baseline variability [83,87]. Using the CCL18 cutoff determined by Prasse et al., another study confirmed that a cutoff of >150 ng/mL for serum CCL18 had the highest accuracy for predicting mortality in IPF patients, consistent with previous reports [83,87]. In general, the value of CCL18 for predicting acute exacerbation of IPF (AE-IPF), a process of rapid deterioration and a major cause of morbidity and mortality in IPF, remains controversial, and correlative research is limited [88,89]; this topic warrants further investigation.

CCL18 is primarily produced by myeloid cells, particularly AMs and follicular dendritic cells. Its expression is induced by Th2-associated cytokines, including IL-4, IL-13 and IL-10, whereas it is downregulated by interferon-γ (IFN-γ) and lipopolysaccharide (LPS) [90]. CCL18 is abundantly produced by AMs in patients with IPF and is defined as a marker of AAMs [91]. Based on the involvement of AAMs in tissue remodeling and fibrosis in chronic inflammatory diseases, CCL18 has been suggested to exert a profibrotic role by inducing the alternatively activated phenotype of AMs in the pathogenesis of IPF. In addition, Antje Prasse et al. [92] documented that AM-produced CCL18 induces collagen synthesis in lung fibroblasts, revealing a positive feedback loop between AMs and lung fibroblasts in which AM-produced CCL18 induces collagen expression and fibroblast-produced collagen increases CCL18 in AMs [92]. The vicious cycle of AMs and fibroblasts via CCL18 was verified by another stud: monomeric collagen type I treatment induces Akt phosphorylation via CD204, shifting AMs to the alternatively activated phenotype and increasing CCL18 production [41]. Interaction between AMs and fibroblasts via CCL18 may perpetuate fibrotic lesions in disease progression (Figure 3). Several groups have investigated the signaling mechanism responsible for CCL18-mediated collagen production in pulmonary fibroblasts [93]. Irina G. Luzina et al. reported that CCL18 activates collagen production in pulmonary fibroblasts through an Sp1-dependent pathway and requires basal Smad3 activity [93]. Unlike another profibrotic chemokine, CCL2, CCL18 signaling is independent of autocrine TGF-β [93]. Moreover, CCL18 intracellular signaling leads to activation of extracellular signal-reduced kinase (ERK)1/2, and pharmacologic inhibition of ERK blocks the effect of CCL18 on collagen production [94]. Such predominant expression of CCL18 in the lungs and its direct stimulation of collagen production independent of TGF-β suggest this chemokine as a potential independent target in strategies to prevent IPF, with fewer adverse effects than TGF-β-targeting therapies [93,94,95].

### 2.9. CCL21

CCL21, also known as 6-Ckine, exodus-2 and secondary lymphoid-tissue chemokine (SLC), is one of the major chemokines predominantly expressed in secondary lymphoid tissues. It exerts effects on target cells by binding to the chemokine receptor CCR7, which is expressed on different immune cells. CCL21-CCR7 interaction essentially contributes to the homing of various subpopulations of T cells and antigen-presenting dendritic cells to the lymph nodes.

The CCL21-CCR7 axis plays important roles in several physiological and pathological processes, such as tissue homeostasis, the inflammatory response and immune surveillance. Several studies have identified involvement of CCR7 and CCL21 in progression of lung fibrogenesis. CCR7 expression is significantly increased in biopsies from patients with IPF compared with that in normal margins [96]. However, no evident differences or clear patterns of CCL21 expression were observed between IPF and normal margin samples, suggesting that this increased expression of CCR7 but not CCL21 within the lung is associated with fibrotic events in IPF. Focal interstitial CCR7 protein is expressed in fibrotic surgical lung biopsies (SLBs) but not in normal margins, and these CCR7-positive (CCR7+) areas in IPF SLBs lack markers for collagen-producing fibrocytes (a distinct population of blood-borne cells exhibiting potent immunostimulatory activities), indicating that focal CCR7 expression may reveal idiopathic injury sites and inappropriate activation of resident fibroblasts [96,97,98]. E.M. Pierce and colleagues [62] elucidated the functional and signaling significance of CCR7 expression in primary fibroblasts obtained from SLB specimens. CCR7 expression is observed in IPF fibroblasts but not in normal fibroblasts. Unlike normal SLB-derived fibroblasts, IPF SLB-derived fibroblasts show significant migratory and proliferative responses when exposed to CCL21 [62]. Moreover, CCR7 activation by CCL21 enhances the capacity of IPF fibroblasts to synthesize chemokines, such as CCL5 [62]. When exposed to CCL21, IPF fibroblasts alter the phosphorylation status of proteins associated with the ERK1/2 mitogen-activated protein kinase pathway [62]. Receptor inhibition with pertussis toxin (PTX) or CCR7 gene silencing with siRNA is able to block CCL21-mediated pathway activation and inhibit functional responses of IPF fibroblasts; thus, CCL21-CCR7 axis-dependent activation of IPF fibroblasts is a promising strategy for IPF treatment [62]. In addition, modulation of the functional properties of fibroblasts by the CCL21-CCR7 axis is restricted in IPF patients and does not occur in healthy cohorts, supporting the crucial roles of CCL21 and CCR7 in IPF pathogenesis [62].

Several studies have verified the in vivo roles of CCL21 and CCR7 in IPF. Researchers injected 1 × 10^6^ primary fibroblasts grown from IPF and other biopsies intravenously into CB-17 severe combined immunodeficiency (SCID)/beige (bg) mice [99]. Compared with introduction of normal fibroblasts, that of IPF fibroblasts caused fibrotic lesions and significantly enhanced hydroxyproline levels in a temporally dependent manner, with effects evident at day 35 and prominent at day 63 after adoptive transfer [99]. Immunoneutralization of CCR7 or CCL21 with monoclonal antibodies abrogated pulmonary remodeling in CB-17-SCID/bg mice that received IPF fibroblasts, highlighting the contribution of CCL21 to IPF pathogenesis [99] and suggesting the possibility of CCL21 as an attractive target in IPF clinical treatment [99].

### 2.10. CCL24

CCL24 (eotaxin-2) is a member of the eotaxin family, which also includes CCL11 (eotaxin-1) and CCL26 (eotaxin-3). It acts as an attractant, inducing trafficking of eosinophils, basophils, neutrophils, and macrophages by binding to CCR3 [100]. Both CCL24 and CCR3 are reported to be involved in IPF. Levels of CCL24 in BALF from patients with IPF are upregulated compared with those in BALF from controls [101]. Additionally, an in vitro study revealed that CCL24 exerts profibrotic effects by stimulating human lung fibroblast proliferation and collagen synthesis [102] and may serve as a marker for differentiating PF from pneumonitis [103]. The efficacy of CM-101, an anti-CCL24 monoclonal antibody, in attenuating inflammation and the pathological process of lung fibrosis was confirmed in a BLM-induced PF model [104]. Compared with mice treated with either PBS or nonspecific immunoglobulin G (IgG), CM-101-exposed mice showed significantly reduced levels of collagen, alleviated histopathological changes and robust attenuation of immune cell infiltration in BALF, supporting a potential therapeutic effect of CM-101 in IPF [104]. This study also established involvement of CCL24 in fibroblast and endothelial cell activation, which are known to participate in IPF pathogenesis, as well as the broad applicability of CM-101 in strategies for inhibiting these activities. Therefore, inhibition of CCL24 or interruption of CCL24-CCR3 signaling is a potential strategy for the treatment of IPF.

## 3. Therapeutic Modulation of CC Chemokines and Receptors in IPF

Based on the evidence that IPF is associated with upregulation of several CC chemokines and chemokine receptors, some studies in mouse models of PF have provided proof-of-principle evidence that targeting the CC chemokine system can reduce disease severity (Table 1). For example, an anti-CCL1 monoclonal antibody ameliorated BLM- and hMSC-induced lung fibrosis in mice, and blockade of CCL3 with a monoclonal antibody or a CCL3-binding protein (evasin-1) attenuated PF in BLM mice. Furthermore, i.v. adoptive transfer of IPF/UIP primary fibroblasts in CB-17-SCID/bg mice induced PF in mice, but administration of a CCL21-blocking monoclonal antibody or an anti-CCR7 monoclonal antibody considerably decreased hydroxyproline levels and attenuated lung interstitial remodeling in these mice. Treatment with an anti-CCL24 monoclonal antibody (CM-101) attenuated immune cell infiltration and reduced collagen deposition in mice with BLM-induced PF, and deletion of CCL2, CCR2, CCL3 or CCR4 induced therapeutic effects in BLM-induced PF mice. However, a phase 2 study conducted in patients with IPF demonstrated that treatment of IPF with carlumab, an immunoglobulin G*_1κ_* monoclonal antibody that specifically binds to and neutralizes human CCL2, failed to provide benefit to patients with progressive disease [45]. Some important issues may help explain the differences in the therapeutic efficacy of antifibrotic agents between the bleomycin model and human IPF. First, the bleomycin model does not recapitulate the histological pattern of human IPF. Second, preclinical trials utilizing mouse models of bleomycin-induced PF often assess the efficacy of a therapeutic compound administered during the inflammatory or early-fibrotic phase. However, in the clinical situation, treatment is initiated during the stage of established fibrosis. Alternative animal models that better reflect human IPF may help to promote clinical translation of chemokine-targeted anti-PF therapies. Collectively, these data suggest that CC chemokines and their receptors are promising targets for new therapies for IPF.

Compared with data from mouse models, data from clinical studies are limited. In a phase 2 trial, carlumab, a human antibody-neutralizing CCL2 [53], failed to produce a benefit in patients with IPF. The reason for this failure is that compensatory mechanisms became overstimulated in response to the inhibition imposed. As the pathogenesis of IPF in humans is undoubtedly more complicated than that in animal models, additional clinical trials are required to assess the potential beneficial effects of targeting CC chemokines and their receptors in IPF.

## 4. Conclusions

Chronic inflammation within the respiratory tract may arise from the combination of diverse environmental insults and genetic susceptibility, and this inflammation underlies the pathogenesis of numerous chronic pulmonary diseases, such as IPF [105]. However, in clinical trials of anti-inflammatory therapy in patients with IPF, triple therapy with the combination of prednisone, azathioprine and N-acetylcysteine was found to be associated with elevated risks of death and hospitalization compared with placebo [106,107,108]. Despite the failure of anti-inflammatory therapy alone for IPF treatment, the inflammatory response may still play an early role in IPF pathogenesis. Recruitment of inflammatory cells is a fundamental process of the early phases of wound healing, and recruitment of immune cells, activation of fibroblasts and excessive deposition of extracellular matrix are common pathophysiological hallmarks of lung fibrosis. Determining the connection between persistent inflammation and abnormal tissue repair may be conducive to IPF treatment.

Apart from their conventional role in recruiting immune cells to the site of inflammation, chemokines also mediate several cellular processes that are specific to IPF remodeling, such as promoting fibroblast trafficking, proliferation and activation. Therefore, chemokines act as a bridge between persistent inflammation and abnormal tissue repair in IPF pathogenesis. Fibroblasts isolated from patients with IPF exhibit distinct characteristics compared with those from healthy donors or patients with other ILDs. As the major effector cells in IPF, fibrotic fibroblasts are hyperresponsive to certain CC chemokines, such as CCL2. In addition, these cells have been proven to serve as a cellular source of some chemoattractants, such as CCL2, CCL7 and CCL8, suggesting a tight link between inflammation and fibroblast activation as well as mutual impact within regional niches. In addition to CC ligands, the multifunctional receptor CCR5 is strongly expressed in fibroblasts from fibrotic lobes, but this expression is less dramatic in ILDs other than IPF [55], further demonstrating the intricate connection of CCLs, CCRs, immune cells and abnormally activated fibroblasts. For instance, the existence of a positive feedback loop between AMs and fibroblasts in patients with IPF is confirmed to perpetuate fibrosis via CCL18 [92]. This vicious cycle highlights the potential significance of cell interactions with CC chemokines as mediators in IPF pathogenesis. Some chemokines have been relatively thoroughly investigated in IPF pathogenesis, including CCL2, CCL18, CCL17 and CCL22. However, mechanistic studies of certain CCLs involved in IPF, such as CCL3, CCL4 and CCL7, are extremely limited. In general, future studies are needed to elucidate the underlying mechanism of these chemokines in the pathogenesis of IPF. Moreover, only a few members of the CC chemokine system have been studied in the context of IPF, and whether other CCLs are involved in IPF pathogenesis requires further exploration.

Highly expressed CC chemokines in serum or BALF, such as CCL2, CCL8, CCL18, CCL17 and CCL22, have been found to have potential diagnostic or prognostic value in IPF. In particular, CCL2 and CCL18 are recognized as diagnostic and prognostic biomarkers for chronic fibrosing ILDs with a progressive phenotype [109]. Nevertheless, inhibition of certain chemokines might not be capable of alleviating PF and attenuating functional impairment because of the complexity of IPF pathogenesis. For example, carlumab, a human monoclonal antibody that specifically binds and neutralizes CCL2, failed to produce benefits in IPF patients [45]. Because of potential compensatory mechanisms by which the body increases free CCL2 levels in response to inhibition, the profibrotic effects of CCL2 cannot be eliminated by neutralizing antibodies alone. Despite a large empirical gap in our knowledge of the chemokine system, studies investigating newer agents that more effectively suppress the CC chemokine system are still worthwhile for IPF treatment.

CCLs exert profibrotic roles in IPF via CCRs and CCL-CCR transduction axes, including CCL21-CCR7, CCL17/CCL22-CCR4 and CCL-CCR5. Inhibition of CCRs or CCL-CCR transduction axes block the function of CCLs. For instance, treatment with PTX or CCR7-specific siRNA is capable of blocking the CCL21-mediated functional responses of IPF fibroblasts [62]. The potential dependence of IPF fibroblasts on certain CCRs and corresponding transduction axes suggests viable therapeutic approaches for disease treatment. However, the immune system in IPF pathogenesis is rather complex. CCR4 deficiency can protect mice from fibrotic responses [73], but a skewed CCR4 to CCR6 CD4+ T-cell ratio in the lung tissue of IPF patients is associated with better lung function [76], suggesting a controversial role for the immune system, potentially distinct fibrotic responses between mice and humans and protective features of specialized T subsets. The complex roles of CCLs and CCLRs in IPF suggest that chemokine-based recruitment of protective T cells can be enhanced while blocking chemokine-based recruitment of detrimental T cells [76]. However, in-depth knowledge of the active mechanisms of CC chemoattractants and receptors in IPF pathogenesis is necessary for context-dependent strategies aiming to treat fibrotic lesions by modulating specialized immune cell subsets.

In summary, extensive in vitro and in vivo investigations have elucidated the crucial roles played by the CC chemokine system in the pathogenesis of IPF (Table 2). The CC chemokine system is thus a promising avenue for therapeutic intervention. Further development of more effective inhibitors of chemokines and their receptors will lead to potential treatment options for IPF. Investigation of CC chemokine-associated mechanisms in IPF pathogenesis is inadequate to date. In-depth exploration of the IPF mechanism and interpretation of the exact roles of chemokines will provide more comprehensive insights into IPF treatment.

## Figures and Tables

**Figure 1 biomolecules-13-00333-f001:**
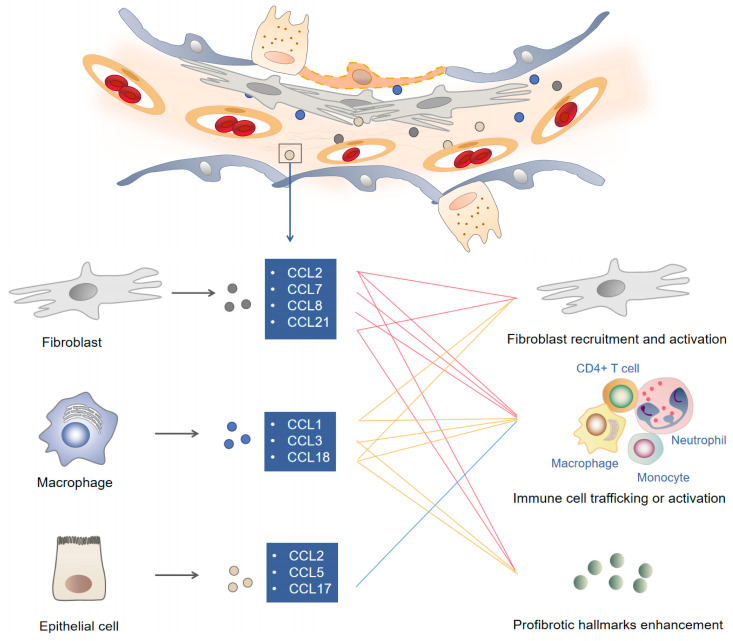
Chemokines and corresponding mechanisms involved in the pathogenesis of idiopathic pulmonary fibrosis. Repetitive stimulation of detrimental initiators leads to long-term pulmonary epithelium impairment. Failure to eliminate initiating factors induces chronic inflammation in the lung. Released chemokines such as CCL1, CCL2 and CCL18 recruit immune cells, impair epithelial cells and activate myofibroblasts to form a profibrotic microenvironment in fibroproliferative lesions. Modulating CCL content favors the alleviation of extracellular matrix deposition and profibrotic hallmark enhancement.

**Figure 2 biomolecules-13-00333-f002:**
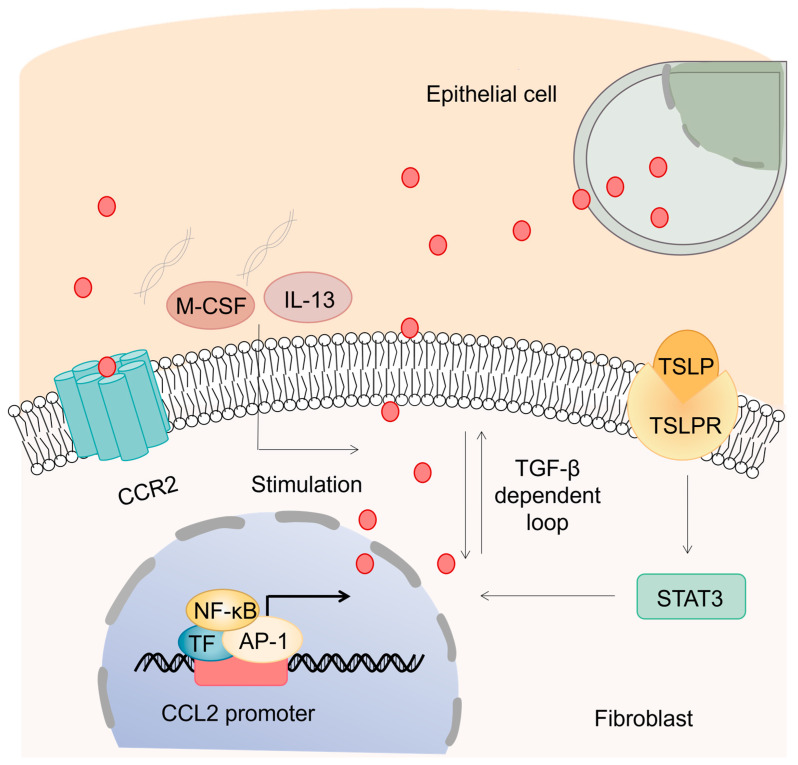
Enhanced CCL2 contributes to myofibroblast activation. CCL2 is mainly secreted by alveolar epithelial cells and pulmonary fibroblasts in IPF. In addition to the enhanced binding of TF to NF-κB and AP-1 elements in the promoter, the TSLP-TSLPR-STAT3 transduction axis and stimulation of M-CSF, IL-13 or collagen type I can also induce augmentation of CCL2 in fibroblasts. CCL2 stimulates collagen synthesis by interacting with CCR2 and upregulating TGF-β. The profibrotic chemoattractant CCL2 participates in the pathogenesis of IPF by activating fibroblasts and inducing collagen deposition.

**Figure 3 biomolecules-13-00333-f003:**
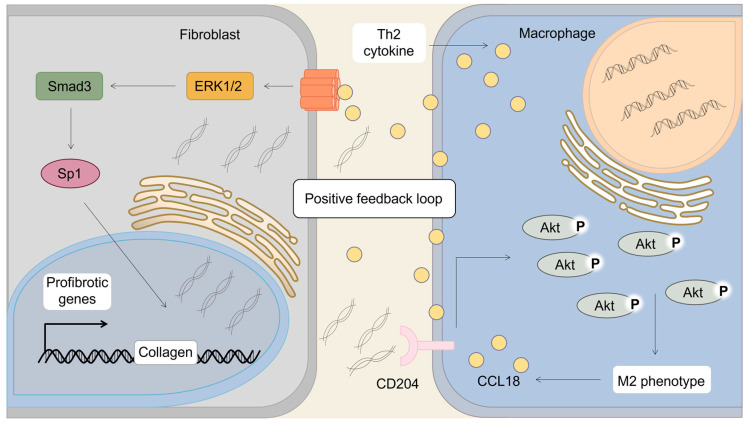
CCL18 mediates the vicious loop of fibroblasts and alveolar macrophages to promote lung fibrosis. CCL18 is primarily produced by alveolar macrophages in IPF. Monomeric collagen type I induces Akt phosphorylation via CD204, shifting alveolar macrophages to the alternatively activated phenotype and augmenting CCL18 production. The enhanced CCL18 interacts with receptors in fibroblasts and induces collagen production via the ERK-Smad3-Sp1 transduction axis. As the effector cells in IPF, fibroblasts form a positive feedback loop with profibrotic alveolar macrophages via CCL18 and collagen. This vicious cycle plays a significant role in IPF.

**Table 1 biomolecules-13-00333-t001:** Targeting chemokine pathways in animal models of PF and human PF.

Target	Model	Species	Inhibition or Genetic Knockout	Outcome	Notes	Refs.
CCL1	BLM-induced PF	Mouse	CCL1 mAb	Effective	Reversed the profibrotic phenotype of fibroblasts; reduced lung collagen deposition; improved lung function; decreased hydroxyproline levels	[20]
	hMSC-induced PF	Mouse	CCL1 mAb	Effective	Suppressed immune cell infiltration; improved lung injury	[19]
CCL2	BLM-induced PF	Mouse	CCL2 deficiency	Effective	Alleviated lung fibrosis; reduced mononuclear phagocyte recruitment; decreased collagen deposition and CTGF expression	[28,29,30]
	IPF	Human	CCL2 mAb (Carlumab)	Not effective	No benefit in lung function	[45]
CCR2	BLM-induced PF; FITC-induced PF	Mouse	CCR2 deficiency	Effective	Decreased collagen deposition; attenuated hydroxyproline content; reduced profibrotic cytokines production	[28,29,30]
CCL3	BLM-induced PF	Mouse	CCL3 mAb	Effective	Reduced pulmonary mononuclear phagocyte accumulation and fibrosis	[50,51,52]
	BLM-induced PF	Mouse	CCL3-binding protein (evasin-1)	Effective	Attenuated pulmonary leukocytes and fibrosis; decreased profibrotic cytokines production	
	BLM-induced PF	Mouse	CCL3 deficiency	Effective	Reduced collagen accumulation; attenuated profibrotic cytokine production	
CCR4	BLM-induced PF	Mouse	CCR4 deficiency	Effective	Decreased mortality; attenuated hydroxyproline contents	[73]
CCL21	i.v. adoptive transfer of either IPF/UIP or NSIP primary fibroblast lines to C.B-17SCID/bg mice	Mouse	CCL21 mAb	Effective	Decreased hydroxyproline levels; attenuated lung interstitial remodeling	[99]
CCR7	i.v. adoptive transfer of either IPF/UIP or NSIP primary fibroblast lines to C.B-17SCID/bg mice	Mouse	CCR7 mAb	Effective	Decreased hydroxyproline levels; attenuated lung interstitial remodeling; reduced whole-lung levels of CCL21	[99]
CCL24	BLM-induced PF	Mouse	CCL24 mAb (CM-101)	Effective	Reduced collagen deposition and fibrosis; attenuated immune cell infiltration	[104]

hMSC: human mesenchymal stem cell; i.v.: intravenous; IPF/UIP: idiopathic pulmonary fibrosis/usual interstitial pneumonia; NSIP: nonspecific interstitial pneumonia; C. B-17SCID/bg mice: C. B-17 severe combined immunodeficiency (SCID)/beige (bg) mice; CTGF: connective tissue growth factor.

**Table 2 biomolecules-13-00333-t002:** Biological characteristics of CC chemokine functions in IPF.

CCLs	Source Cells	Target Cells	Functions	Refs.
CCL1	AMφ	Monocyte, Mφ, Lymphocyte, Fibroblast	Immune cell trafficking; Fibroblast recruitment; Fibroblast activation	[19,20,91,110,111,112]
CCL2	AEC, Fibroblast	Monocyte, Mφ, Fibroblast	Mφ and fibroblast activation; Monocyte recruitment; TGFβ1 upregulation; Profibrotic hallmarks enhancement	[27,32,36,37,38]
CCL3	Mφ, Granulocyte	Granulocyte, T-lymphocyte, Mφ	Mφ and fibrocyte recruitment; Profibrotic factors enhancement	[25,47,50,51,52,113,114]
CCL4	Elusive	Granulocyte, T-lymphocyte, Mφ	Immune cell recruitment	[47,114,115]
CCL5	Epithelial cell	Granulocyte, Lymphocyte, Mφ, Fibroblast	Elusive	[53,54,55,116]
CCL7	Fibroblast	Granulocyte, Monocyte, Lymphocyte, Dendritic cell	Immune cell activation	[55,57]
CCL8	Fibroblast	Granulocyte, Monocyte, Lymphocyte	Immune cell activation	[57,64]
CCL17	Epithelial cell	Mφ, Lymphocyte	Mφ and lymphocyte activation	[49,71,72,73,74,75,76]
CCL18	AMφ	Fibroblast	Marker of alternative Mφ activation; Fibroblast activation; Collagen enhancement	[83,90,91,92,117,118,119,120,121,122]
CCL21	Fibroblast	Fibroblast	Fibroblast activation; Enhancement of other chemokines	[62,99]
CCL22	Elusive	AMφ	Mφ and lymphocyte activation	[49,71,72,73,74,75,76]
CCL24	Elusive	Granulocyte, Fibroblast	Fibroblast activation	[100,102,104,123,124]

Mφ: macrophage; AMφ: alveolar macrophage; AEC: alveolar epithelial cell.

## Data Availability

Not applicable.

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
