# Peer review of "CC Chemokines in Idiopathic Pulmonary Fibrosis: Pathogenic Role and Therapeutic Potential"

_biomolecules, 2023, doi:10.3390/biom13020333_

Round 1

Reviewer 1 Report

Here, Liu and coworkers provide a review article summarizing the individual role of CC chemokines in idiopathic pulmonary fibrosis (IPF). In fact and in line with most primary research articles the role of CC chemokines in IPF is mainly correlative and not causative. Moreover, a lot of information described here for individual chemokine are outdated and wrong. In addition, the function of individual chemokines on target cells do not match with the corresponding chemokine receptor expression pattern on the described target cells. The errors are too numerous to be listed individually here. But PMID 24218476 and 29637711 provide excellent and comprehensive description of chemokine and receptor patterns.

Major concerns:

-          Chapter 2: General introduction to each chemokine includes outdated and wrong information. Similarly, chemokine receptor expression in different leukocyte subsets are occasionally also outdated. PMID 24218476 and 29637711 provide excellent and comprehensive description of chemokine and receptor patterns.
e.g. CCL1 is – most importantly for this study - also expressed by endothelial cells and activated monocytes!
and its cognate receptor CCR8 is predominantly expressed on Th2 T cells, but not on B cells or DCs. Hence CCL1 attracts primarily Th2 cells and not monocytes, immature B cells and DCs as proclaimed by Liu and colleagues.

The same is true for many other chemokines and their target cells.

-          Another such major conflict is the authors claim that CCL3 levels in IPF patients correlate significantly with the total number of neutrophils and eosinophils (lane 207ff). CCL3 binds to CCR1 and CCR5, whereas eosinophils solely express CCR3 – the receptor for CCL11.

-          Lane 248 ff. Fibroblasts do not express CCR5, but infiltrated leukocytes in IPF do!

-          Lane 438ff. CCR7 is also not expressed by fibroblasts, but by infiltrating adaptive immune cells!

Title for chapter 2.4 should read CCL3 and CCL5 (not CCL4!)

Author Response

Here, Liu and coworkers provide a review article summarizing the individual role of CC chemokines in idiopathic pulmonary fibrosis (IPF). In fact and in line with most primary research articles the role of CC chemokines in IPF is mainly correlative and not causative. Moreover, a lot of information described here for individual chemokine are outdated and wrong. In addition, the function of individual chemokines on target cells do not match with the corresponding chemokine receptor expression pattern on the described target cells. The errors are too numerous to be listed individually here. But PMID 24218476 and 29637711 provide excellent and comprehensive description of chemokine and receptor patterns.

We thank for Reviewer #1 for raising these important questions.

  • For Reviewer #1’s opinion that the role of CC chemokines in idiopathic pulmonary fibrosis (IPF) is mainly correlative and not causative, we think that our study including others have verified the causative effects of several chemokines on pulmonary fibrosis (PF). For example, we found that elevated levels of CCL1 are sufficient to promote fibrosis in lung tissue, and antibody blockade of CCL1 ameliorates PF pathology. In addition, deficiency of CCL2, CCR2, CCL3, CCR4, et, al. has been reported to block the BLM-induced lung pathology. All these data suggest that the roles of several reported chemokines in IPF may be not only correlative but causative.
  • We thank for Reviewer #1’s comments about the outdated and wrong information for each chemokine in chapter 2. We carefully checked them and provided evidence for most questions the Reviewer raised, which can be seen in our “point by point response”. We think that the expressing cells of chemokines and their receptors vary under different disease conditions. Moreover, we believe that the expressing cells of chemokines and their receptors are also different between health and disease states.

Major concerns:

  1. Chapter 2: General introduction to each chemokine includes outdated and wrong information. Similarly, chemokine receptor expression in different leukocyte subsets are occasionally also outdated. PMID 24218476 and 29637711 provide excellent and comprehensive description of chemokine and receptor patterns.

e.g. CCL1 is – most importantly for this study - also expressed by endothelial cells and activated monocytes! and its cognate receptor CCR8 is predominantly expressed on Th2 T cells, but not on B cells or DCs. Hence CCL1 attracts primarily Th2 cells and not monocytes, immature B cells and DCs as proclaimed by Liu and colleagues.

The same is true for many other chemokines and their target cells.

Re: Thank you for your suggestion, we have cited those two papers you recommend (PMID 24218476 and 29637711) and carefully revised the description of chemokine and receptor patterns according to those papers.

We agree with the reviewer that CCL1 is also expressed by endothelial cells and activated monocytes, and we have revised its expressing cells in the revised manuscript.

We also agree that CCL1 can attracts Th2 cells. However, its chemotactic effect on monocytes (PMID: 1557400, PMID: 10942748 and PMID: 9207005), immature B cells (PMID: 9207005 and PMID: 9211859) and DCs (PMID: 15814739 and PMID: 30170811) were also demonstrated in various literatures.

jM D Miller, M S Krangel. The human cytokine I-309 is a monocyte chemoattractant. Proc Natl Acad Sci U S A. 1992 Apr 1;89(7):2950-4. doi: 10.1073/pnas.89.7.2950.

kN S Haque, X Zhang, D L French, J Li, M Poon, J T Fallon, B R Gabel, M B Taubman, M Koschinsky, P C Harpel. CC chemokine I-309 is the principal monocyte chemoattractant induced by apolipoprotein(a) in human vascular endothelial cells. Circulation. 2000 Aug 15;102(7):786-92. doi: 10.1161/01.cir.102.7.786.

lH L Tiffany, L L Lautens, J L Gao, J Pease, M Locati, C Combadiere, W Modi, T I Bonner, P M Murphy. Identification of CCR8: a human monocyte and thymus receptor for the CC chemokine I-309. J Exp Med. 1997 Jul 7;186(1):165-70. doi: 10.1084/jem.186.1.165.

mR S Roos, M Loetscher, D F Legler, I Clark-Lewis, M Baggiolini, B Moser. Identification of CCR8, the receptor for the human CC chemokine I-309. J Biol Chem. 1997 Jul 11;272(28):17251-4. doi: 10.1074/jbc.272.28.17251

nMichael Gombert, Marie-Caroline Dieu, Franziska Winterberg, Erich Bünemann, Robert C Kubitza, Ludivine Da Cunha, et al. CCL1-CCR8 interactions: an axis mediating the recruitment of T cells and Langerhans-type dendritic cells to sites of atopic skin inflammation. J Immunol. 2005 Apr 15;174(8):5082-91. doi: 10.4049/jimmunol.174.8.5082.

oCaroline L. Sokol, Ryan B. Camire, Michael C. Jones, and Andrew D. Luster. The chemokine receptor CCR8 promotes the migration of dendritic cells into the lymph node parenchyma to initiate the allergic immune response. Immunity. 2018 Sep 18; 49(3): 449–463.e6. doi: 10.1016/j.immuni.2018.07.012.

  1. Another such major conflict is the authors claim that CCL3 levels in IPF patients correlate significantly with the total number of neutrophils and eosinophils (lane 207ff). CCL3 binds to CCR1 and CCR5, whereas eosinophils solely express CCR3 – the receptor for CCL11.

Re: Thank you for your comments. The description that “CCL3 levels in IPF patients correlate significantly with the total number of neutrophils and eosinophils” was reported by Ali Emad in 2007 (PMID: 17720292). They found that among the group B (bronchoalveolar lavage fluid of IPF patients), the level of MIP-α (an aliase for CCL3) was significantly correlated only with the absolute number and the percentage of neutrophils (r = 0.65, p = 0.008; r = 0.52, p = 0.04; respectively) and the number and the percentage of eosinophils (r = 0.51, p = 0.04; r = 0.56, p = 0.02; respectively).

Ali Emad, Vahid Emad. Elevated levels of MCP-1, MIP-alpha and MIP-1 beta in the bronchoalveolar lavage (BAL) fluid of patients with mustard gas-induced pulmonary fibrosis. Toxicology. 2007 Oct 30;240(1-2):60-9. doi: 10.1016/j.tox.2007.07.014.

  1. Lane 248 ff. Fibroblasts do not express CCR5, but infiltrated leukocytes in IPF do!

Re: Although CCR5 is noted to be expressed on various cell types, including T cells, macrophages, dendritic cells, eosinophils, and microglia. Researchers have detected the presence of CCR5 in the context of UIP (also referred to as IPF if the disease is idiopathic) using immunocytochemical analysis. They confrimed that upper- and lower-lobe fibroblasts from patients with UIP exhibited very strong expression of CCR5, and this receptor was observed on the majority of the fibroblasts in culture (PMID 15191918) .

Esther S. Choi, Claudia Jakubzick, Kristin J. Carpenter, Steven L. Kunkel, Holly Evanoff, Fernando J. Martinez, Kevin R. Flaherty, Galen B. Toews, Thomas V. Colby, Ella A. Kazerooni, Barry H. Gross, William D. Travis, and Cory M. Hogaboam. Enhanced Monocyte Chemoattractant Protein-3/CC Chemokine Ligand-7 in Usual Interstitial Pneumonia. Am J Respir Crit Care Med. 2004 Sep 1;170(5):508-15. doi: 10.1164/rccm.200401-002OC.

  1. Lane 438ff. CCR7 is also not expressed by fibroblasts, but by infiltrating adaptive immune cells!

Re: We thank this reviewer’s comments. CCR7 is a receptor once thought to be restricted to haematopoietic cells. However, several studies have revealed that in contrast to primary normal fibroblasts, nearly 100% of fibroblasts cultured from histologically proven IPF/UIP specimens expressed CCR7 (PMID17331965 and PMID 16394278). We believe that the expressing cells of chemokines and their receptors vary under different disease condition.

j E M Pierce, K Carpenter, C Jakubzick, S L Kunkel, H Evanoff, K R Flaherty, F J Martinez, G B Toews, C M Hogaboam. Idiopathic pulmonary fibrosis fibroblasts migrate and proliferate to CC chemokine ligand 21. Eur Respir J. 2007 Jun;29(6):1082-93. doi: 10.1183/09031936.00122806.

k E S Choi, E M Pierce, C Jakubzick, K J Carpenter, S L Kunkel, H Evanoff, F J Martinez, K R Flaherty, B B Moore, G B Toews, T V Colby, E A Kazerooni, B H Gross, W D Travis, and C M Hogaboam. Focal interstitial CC chemokine receptor 7 (CCR7) expression in idiopathic interstitial pneumonia. J Clin Pathol. 2006 Jan; 59(1): 28–39. doi: 10.1136/jcp.2005.026872.

l Elizabeth M Pierce, Kristin Carpenter, Claudia Jakubzick, Steven L Kunkel, Kevin R Flaherty, Fernando J Martinez, Cory M Hogaboam. Therapeutic targeting of CC ligand 21 or CC chemokine receptor 7 abrogates pulmonary fibrosis induced by the adoptive transfer of human pulmonary fibroblasts to immunodeficient mice. Am J Pathol. 2007 Apr;170(4):1152-64. doi: 10.2353/ajpath.2007.060649.

  1. Title for chapter 2.4 should read CCL3 and CCL5 (not CCL4!)

Re: We thank the reviewer for pointing out this mistake. We have replaced “CCL3 and CCL4” with “CCL5” in title for chapter 2.4.

Reviewer 2 Report

In this article, Liu et. al. have reviewed the roles of different CC chemokines in the pathogenesis of IPF, and discussed their potential use as biomarkers or therapeutic targets. It is a well-written review with systematic organization of information. Specifically, the 3 figures and the table that the authors have presented will be helpful to the readers in comprehension of the concepts. I have a few comments which are listed below:

1. While IPF was historically believed to be a disease driven by chronic inflammation, repeated failures of anti-inflammatory therapies have decreased confidence in this concept in the field over the years. While the authors do allude to this later in the review article, it might be helpful to mention this upfront in the introduction section.

2. In a relatively recent paper (2019), in the American Journal of Respiratory Cell and Molecular Biology, Yang et. al. reported that CCL2/12 derived from injured alveolar epithelial cells promotes lung fibrosis. It may be helpful to cite that paper.

“Yang J, Agarwal M, Ling S, Teitz-Tennenbaum S, Zemans RL, Osterholzer JJ, Sisson TH, Kim KK. Diverse Injury Pathways Induce Alveolar Epithelial Cell CCL2/12, Which Promotes Lung Fibrosis. Am J Respir Cell Mol Biol. 2020 May;62(5):622-632. doi: 10.1165/rcmb.2019-0297OC. PMID: 31922885; PMCID: PMC7193786.”

3. In Figure 1, for “immunocyte trafficking and activation”, can the authors define the different immunocytes that they have graphically depicted?

4. Section 2.4 – Replace “CCL3 and CCL4” by “CCL5”

5. Can they add further discussion on the differences between the bleomycin model and human IPF such that while targeting chemokine pathways in the mouse model is effective in multiple studies, that is not the case in human IPF (as depicted in their table 1)?

Author Response

In this article, Liu et. al. have reviewed the roles of different CC chemokines in the pathogenesis of IPF, and discussed their potential use as biomarkers or therapeutic targets. It is a well-written review with systematic organization of information. Specifically, the 3 figures and the table that the authors have presented will be helpful to the readers in comprehension of the concepts.

Re: We thank for the reviewer’s enthusiastic help and professional guidance that make our manuscript gaining a significant improvement.

I have a few comments which are listed below:

  1. While IPF was historically believed to be a disease driven by chronic inflammation, repeated failures of anti-inflammatory therapies have decreased confidence in this concept in the field over the years. While the authors do allude to this later in the review article, it might be helpful to mention this upfront in the introduction section.

Re: Thank you for your suggestion. Failures of anti-inflammatory therapies and current therapies for IPF in clinic were described in the introduction section.

  1. In a relatively recent paper (2019), in the American Journal of Respiratory Cell and Molecular Biology, Yang et. al. reported that CCL2/12 derived from injured alveolar epithelial cells promotes lung fibrosis. It may be helpful to cite that paper.

“Yang J, Agarwal M, Ling S, Teitz-Tennenbaum S, Zemans RL, Osterholzer JJ, Sisson TH, Kim KK. Diverse Injury Pathways Induce Alveolar Epithelial Cell CCL2/12, Which Promotes Lung Fibrosis. Am J Respir Cell Mol Biol. 2020 May;62(5):622-632. doi: 10.1165/rcmb.2019-0297OC. PMID: 31922885; PMCID: PMC7193786.”

Re: Following your suggestion, we have cited this paper (Reference No.38).

  1. In Figure 1, for “immunocyte trafficking and activation”, can the authors define the different immunocytes that they have graphically depicted?

Re: Thank you for your suggestion, the different immunocytes in Figure 1 have been defined beside the depicted graph.

  1. Section 2.4 – Replace “CCL3 and CCL4” by “CCL5”

Re: We thank the reviewer for pointing out this mistake. We have replaced “CCL3 and CCL4” with “CCL5” in Section 2.4.

  1. Can they add further discussion on the differences between the bleomycin model and human IPF such that while targeting chemokine pathways in the mouse model is effective in multiple studies, that is not the case in human IPF (as depicted in their table 1)?

Re: Thank you for your suggestion. We have discussed the differences of the therapeutic efficacy of chemokine-targeted therapies between the bleomycin model and human IPF in part “3. Therapeutic modulation of CC chemokines and receptors in IPF”.
